# The Secreted Protein MoHrip1 Is Necessary for the Virulence of *Magnaporthe oryzae*

**DOI:** 10.3390/ijms20071643

**Published:** 2019-04-02

**Authors:** Hai-Zhen Nie, Lin Zhang, Hui-Qian Zhuang, Wen-Jiong Shi, Xiu-Fen Yang, De-Wen Qiu, Hong-Mei Zeng

**Affiliations:** The State Key Laboratory for Biology of Plant Diseases and Insect Pests, Institute of Plant Protection, Chinese Academy of Agricultural Sciences, Beijing 100081, China; niehaizhen880701@163.com (H.-Z.N.); 82101171169@caas.cn (L.Z.); daqian12@sina.cn (H.-Q.Z.); 82101176113@caas.cn (W.-J.S.); yangxiufen@caas.cn (X.-F.Y.); qiudewen@caas.cn (D.-W.Q.)

**Keywords:** *Magnaporthe oryzae*, MoHrip1, expression pattern, virulence

## Abstract

Secreted effectors from *Magnaporthe oryzae* play critical roles in the interaction with rice to facilitate fungal infection and disease development. *M. oryzae*-secreted protein MoHrip1 can improve plant defense as an elicitor in vitro, however, its biological function in fungal infection is not clear. In this study, we found that the expression of *mohrip1* was significantly induced in the stages of fungal penetration and colonization. Although dispensable for the growth and conidiation, MoHrip1 was necessary for the full virulence of *M. oryzae*. Deletion of *mohrip1* remarkably compromised fungal virulence on rice seedlings and even on rice leaves with wounds. Rice sheath inoculation assay further demonstrated the defects of *mohrip1*-deleted mutants on penetration and proliferation in rice cells. Additionally, compared with WT and complementation strain, the inoculation of *mohrip1*-deleted mutants induced a higher expression of specific defense related genes and a higher production of specific defensive compounds in rice leaves. These data collectively indicated that MoHrip1 is necessary for fungal penetration and invasive expansion, and further full virulence of rice blast fungus.

## 1. Introduction

Biotrophic or hemibiotrophic pathogenic fungi often feed on living host tissues for the whole life cycle or early infection stages. In the process of pathogen infection, many strategies are deployed to disturb host immunity or inactivate host defense responses, among which, secreted effectors play a key role [1,2].

*Magnaporthe oryzae*, a synonym of *Pyricularia oryzae* [3], the causal agent of blast disease, significantly constrains rice production worldwide [4]. In a typical disease cycle, infections initiate when the conidia of *M. oryzae* land on the surface of rice leaves and germinate to form germ tubes, which then differentiate into melanized appressoria. With the enormous turgor pressure and physical force, penetration pegs, developing from these specialized appressoria, breach the host cuticle and subsequently differentiate into biotrophic invasive hyphae (IH) [5,6,7,8]. The thin tubular primary IH develop into bulbous pseudohyphae and then filamentous hyphae, filling the infected cells [6,9]. In this invasive process, a myriad of effectors, such as AvrPiz-t [10] and biotrophy-associated secreted (BAS) proteins [11], are secreted to the pathogen-host interface by the fungal secretion machinery, which locates adjacent to the bulbous pseudohyphae [1,9,12,13]. These secreted effectors mainly function to alter host cellular defense processes and facilitate further infections [14].

In recent years, machine-learning tools, which incorporate common features of fungal effectors discovered to date (including protein size, amino acid content, charge, evidence for diversifying selection, etc.), have been adopted for the prediction of putative effectors from filamentous pathogens, which often possess huge effector repertoires comprising hundreds of sequence-unrelated small secreted proteins, for example, the newly identified family termed as MAX-effector (*M. oryzae* Avrs and ToxB) [2,12]. But in general, data on the molecular function of these effectors are sparse. As one member of MAX-effector family, AVR-Pia has been identified as an avirulence factor to trigger hypersensitive response (HR) and activate innate immunity in rice cultivars containing the *Pia* R gene [2,15,16,17]. In the initial invasive stage, AVR-Pia is induced at the onset of appressorial differentiation and recognized by rice NLR protein RGA5 through direct binding to induce the downstream signal transduction [18,19,20]. Moreover, with the analysis of interaction transcriptome, PWL2 [21], also considered as an avirulence factor, and other 58 candidate effectors have been identified during *M. oryzae* invading the living rice cells [11]. Among these effectors, BAS1 functions mainly as a virulence factor to promote blast fungus growth, sporulation and virulence in planta [22]. The same as BAS1, other effectors, MC69 [23] and Rbf1 [9], have been successively identified as virulence factors to facilitate fungal invasion. These secreted effectors often act together, thus the deletion of a single effector gene generally compromises the fungal virulence but has no all-or-none effect [23].

MoHrip1, first identified from the culture filtrate of *M. oryzae*, is a HR inducing protein elicitor. The *mohrip1* gene (GenBank accession No. JQ231215.1) encodes a small protein consisting of 142 amino acids with the first 16 amino acids as a signal peptide. In our previous research, MoHrip1 had been shown to induce the early events of defense responses in tobacco and to enhance systemic resistance of rice to blast fungus in vitro [24]. Furthermore, if ectopic expressed in rice, MoHrip1 can also improve plant resistance to blast disease and tolerance to drought [25]. But as a secreted effector, the intrinsic function of MoHrip1 in *M. oryzae* and especially in the process of *M. oryzae* infecting rice is not clear. In this study, we demonstrated that the expression of *mohrip1* was induced during *M. oryzae* infection. More importantly, MoHrip1 was demonstrated to facilitate fungal penetration and subsequent proliferation by the suppression of rice immunity, which guaranteed the full virulence of *M. oryzae*. Our data corroborate the function of most effectors from plant pathogens to facilitate infection and pathogenesis.

## 2. Results

### 2.1. The Mohrip1 Is Highly Expressed During Fungal Penetration and Colonization

MoHrip1 was originally isolated from the culture filtrate of *M. oryzae* as a protein elicitor [24]. Thus, it is expressed and secreted at least in the vegetative growth stage. To investigate the expression pattern of *mohrip1* in *M. oryzae* at different developmental and infection stages, quantitative reverse transcription polymerase chain reaction (qRT-PCR) assays were conducted. As shown in Figure 1, the transcriptional levels of *mohrip1* in the saprophytic stages, either conidia obtained from oat agar (OA) plate or mycelia from liquid complete medium (CM), were similar and significantly lower than those in the infection stages. After the inoculation of *M. oryzae* conidia on rice seedlings, the abundance of *mohrip1* transcripts demonstrated a tendency of increasing at first and then decreasing to the normal level as in the axenic mycelia or conidia. Compared to the liquid-cultured mycelia, mRNA accumulation increased almost sevenfold at the first and second day post inoculation (dpi), which respectively coincided with fungal penetration and colonization stages. Consequently, the expression pattern of *mohrip1* indicated its significance in *M. oryzae* infection.

### 2.2. MoHrip1 Is Dispensable for Fungal Growth and Development In Vitro

MoHrip1, whether in native or recombinant form, induces HR in tobacco and triggers intensive defense responses in rice in vitro [24]. But its intrinsic function in *M. oryzae* and especially in its interaction with rice is still elusive. Basing on this problem, targeted gene deletion was conducted by replacing *mohrip1* with hygromycin B resistance gene *HPT* according to the principle of homologous recombination (Figure 2A). Two mutants (*Δmohrip1-10* and *Δmohrip1-30*) were selected by resistance to hygromycin B and further confirmed by PCR and Southern Blot analysis (Figure 2B).

The two independent deletion mutants, *Δmohrip1-10* and *Δmohrip1-30*, displayed identical phenotypes in vitro, thus *Δmohrip1-10* was selected as a representative in some of the following bioassays. The mutants were not distinct from WT in colony growth, production and morphology of conidia and appressoria (Figure 3), indicating that MoHrip1 was unnecessary for fungal saprophytic growth and conidiation. This was similar to some other identified effectors from *M. oryzae* [9,23]. As for the estimate of virulence on rice, the mutants demonstrated remarkable defects (Figure 4). To further test if the reduced virulence was exclusively due to the deletion of *mohrip1*, a complementation strain (*mohrip1-10-com*) basing on *Δmohrip1-10* was selected (Figure 2C) and exhibited normal growth, conidiation and infection as WT (Figure 3 and Figure 4). Subsequently, the virulence of these mutants was analyzed in detail.

### 2.3. MoHrip1 Is Required for the Full Virulence of M. oryzae

Adopting the classic spraying inoculation method, we inoculated the conidia of *Δmohrip1-10*, *Δmohrip1-30*, *mohrip1-10-com* and WT on 3-week-old rice seedlings. The disease symptoms were surveyed at 7 dpi. Referring to the lesion types established by Valent et al. [26], many large eyespot lesions were formed and considerably overlapped together on rice leaves inoculated with conidial suspension of WT. While for the deletion mutants, just a few dark brown necrotic spots scattered on the infected leaves (Figure 4A). To further confirm the observed symptoms quantitatively, disease status was rated according to the international specification for rice blast disease. Consistently, the obtained disease index of WT was significantly larger than those of the deletion mutants (Figure 4B). However, the impaired virulence could be restored by the complementation of the *mohrip1* gene with its native promoter and terminator. Consequently, MoHrip1 functioned as an important virulence factor during *M. oryzae* infection.

To further confirm the result of the spraying inoculation bioassay, another method, punch inoculation, was conducted on 6 to 8-week-old rice plants [27]. Disease status was investigated at 12 dpi. Again, WT and complementation strain *mohrip1-10-com* produced larger lesions on rice leaves than those caused by the deletion mutants (Figure 4C). In addition, the relative lesion area and fungal biomass in these lesions were also examined and corroborated the symptoms quantitatively. The relative lesion area, calculated using the Adobe Photoshop software, and the relative fungal biomass, represented by the transcription ratio of *M. oryzae Pot2* gene versus rice *Ubiquitin* gene, were both significantly smaller in the rice plants inoculated with the *mohrip1*-deleted mutants than WT and complementation strain (Figure 4D,E). All these data coincided with the results of spraying inoculation bioassay and demonstrated the significance of MoHrip1 for the full virulence of *M. oryzae*.

### 2.4. MoHrip1 Is Important for the Penetration and Proliferation of M. oryzae

Generally, deficiencies in penetration and infectious growth are the two main limiting factors for the impaired virulence of pathogens. In most researches for *M. oryzae*, the penetration ratio in the first 24 hours post inoculation (hpi) is often taken as an indicator of the ability to penetrate [28,29]. Consequently, an inoculation assay using rice sheaths was conducted to observe the infection process under a microscope. Conidial suspensions (5 × 10^4^ conidia mL^−1^ in 0.2% Tween 20) of *Δmohrip1-10*, *mohrip1-10-com* and WT were respectively inoculated on the detached rice sheaths placed in a moist box. According to the four grades we divided in another research [30], the percentages of different infection levels were calculated. At 12 hpi, the percentages of formed appressoria reached approximately 80% for all the strains, revealing that MoHrip1 was dispensable for the formation of invasive structure. Subsequently at 24 hpi, as shown in Figure 5A, nearly 70% of the appressoria of *Δmohrip1-10* didn’t form the primary IH. While for the complementation strain *mohrip1-10-com* and WT, the percentages of appressoria developing IH reached approximately 60%. Therefore, the results indicated that MoHrip1 played an important role in the penetration of *M. oryzae*.

To further explore whether the proliferation of *Δmohrip1-10* was restricted in the initial infected rice cell or not, sheath inoculation assay was continually monitored to 36 hpi. As shown in Figure 5B, IH of WT spread freely in the initial and adjacent rice cells, while for *Δmohrip1-10*, most of the penetrated hyphae were restricted in the initial invaded cell. And this defect could be restored by the complementation of *mohrip1* gene. According to the statistical results, almost 60% of the IH came to Level 3, and the proportions of Level 2 and Level 1 were gradually decreased to about 25% and 15% for WT and *mohrip1-10-com*. By contrast, approximately 43% and 31% IH of *Δmohrip1-10* were limited in the status of Level 1 and Level 2. While for Level 3, the proportion was just 12% (Figure 5B). Consequently, MoHrip1 was also crucial for the infectious growth and spread of IH in the infected rice cells. In conclusion, MoHrip1 was important for fungal penetration and subsequent proliferation, and thus the full virulence of *M. oryzae*.

### 2.5. MoHrip1 Suppresses the Activation of Plant Immunity

In the process of *M. oryzae* infecting rice, the pathogen deploys different strategies to contribute to its virulence. Multiple defense responses in rice are also induced correspondingly to defend [31,32], among which includes the up-regulated expression of certain defense related genes, which play important roles in the restriction of fungal penetration and proliferation [9,33]. To explore whether the compromised virulence of *Δmohrip1* mutants was partially due to their defect in suppression of plant defense responses, qRT-PCR assays were conducted. The obtained results demonstrated that the transcription of a subset of defense related genes differed significantly at 3 dpi relying on whether the inoculated strain was *mohrip1* deleted or not. As shown in Figure 6, the expression levels of *PBZ1*, *PAL* and *PR1a*, were significantly higher in rice plants inoculated with *Δmohrip1-10* than those with WT. While for *PR5*, the opposite tendency was obtained. In addition, the transcripts of four genes related to the biosynthesis of phytoalexins, especially oryzalexin, were also estimated. Consistently, their expression also demonstrated an obvious increase in rice leaves inoculated with *Δmohrip1-10* compared with WT (Figure 6).

Biosynthesis of sakuranetin, the only known flavonoid phytoalexin in rice to date, depends significantly on *OsNOMT* gene, which encodes a naringenin 7-*O*-methyltransferase [34,35]. After inoculation with *Δmohrip1-10*, the transcripts of *OsNOMT* were significantly higher than those in WT infected leaves (Figure 7A). To further confirm the expression of *OsNOMT*, we estimated the contents of sakuranetin in rice leaves inoculated with *M. oryzae* strains at 3 dpi. Consistently, *Δmohrip1-10* induced significantly higher accumulation of sakuranetin (Figure 7B). Furthermore, we also measured the content of another phytoalexin momilactone A. Consistently, its accumulation in rice leaves inoculated with *Δmohrip1-10* was also significantly higher than WT (Figure 7C). Therefore, we concluded that MoHrip1 suppressed the accumulation of certain phytoalexins. In addition, we also determined the contents of total phenols and flavonoids in rice leaves. Coincided with the production of sakuranetin and momilactone A, the contents of these defensive compounds were also improved in *Δmohrip1-10*-treated rice leaves, although no significance was detected in the contents of total phenols between *Δmohrip1-10* and WT (Figure 7D,E). Altogether, the result demonstrated that MoHrip1 inhibited the expression of specific defense related genes and the biosynthesis of specific defensive compounds, resulting in the inactivation of rice plant immunity.

## 3. Discussion

In this study, we mainly characterized the biological function of MoHrip1 in *M. oryzae* and especially the infective interaction with rice. The obtained results demonstrated that the expression of *mohrip1* was significantly improved during fungal infection. What’s more, the significantly induced MoHrip1 played a critical role in fungal penetration and proliferation by suppression of rice immunity, and this was the main cause for the compromised virulence of *mohrip1*-deleted mutants.

With the prosperous development of pathogen-plant interaction, numerous effectors have been identified and shown to be important for the virulence of fungal pathogens and modulation of plant immunity [36,37]. Generally, many of these candidate effectors are significantly induced during the infection process [11,38]. In our study, we also investigated the expression pattern of *mohrip1* by qRT-PCR and found that its transcription was obviously up-regulated in the first 2 dpi, followed by a gradual decrease for up to five days to the normal level as in axenic mycelia or conidia (Figure 1). This kind of expression pattern was similar to many other effectors from *M. oryzae*. *PWL2* and other 58 putative effector genes have been found to be up-regulated at least tenfold during plant infection by an analysis of interaction transcriptome [11]. Additionally, Sornkom et al. [18] have analyzed in detail the expression pattern of *Avr-Pia*. They divided the time course post inoculation into three stages: 0 to 24 hpi, 24 to 36 hpi and after 36 hpi respectively representing the stages of appressorial differentiation, penetration and colonization. Accordingly, the obvious induction of *mohrip1* extremely coincided with the penetration and colonization stages. Therefore, it is very likely that MoHrip1 plays a crucial role in the invasion and proliferation of *M. oryzae*.

To investigate the biological function of MoHrip1 in *M. oryzae*, deletion and complementation mutants were constructed (Figure 2). In the phenotypic measurement of growth and development, we didn’t find any difference between the deletion mutants and WT in mycelial growth, conidiation, and formation of appressoria (Figure 3). This was similar to many other effectors from *M. oryzae*, for example, RBF1 [9] and MC69 [23]. However, the virulence of deletion mutants on rice seedlings (spraying inoculation) and even on rice leaves with wounds (punch inoculation) was largely compromised (Figure 4). Therefore, we speculated that the compromised virulence of *mohrip1*-deleted mutants might partially result from their defect in the hyphal spread after penetration. This hypothesis was further corroborated by the results from rice sheath inoculation assay (Figure 5). Besides, the assay also demonstrated the significantly compromised penetration ability of the deletion mutants. These results collectively indicated the importance of MoHrip1 in fungal penetration and colonization, and this was coherent to the expression pattern of *mohrip1* (Figure 1).

Our previous research has proved the function of purified MoHrip1 as an exogenous inducer to elicit the early events of defense response in tobacco, such as hydrogen peroxide production, callose deposition, and alkalization of the extracellular medium [24]. In rice, the MoHrip1-treated seedlings possess enhanced systemic resistance to blast fungus, and the results have been further confirmed by the transcriptional profiling [24,39]. Additionally, over-expression of the *mohrip1* gene in rice also significantly enhances plant resistance to blast fungus [25]. Gathering all these results together, it may seem puzzling that MoHrip1 acts both as a virulence factor facilitating fungal infection and an elicitor activating plant immune responses. However, many other effectors have also been proved functioning in the similar way. MSP1, one member from the snodprot1 family, possesses significant phytotoxic properties in the infective interaction between *M. grisea* and rice. But the purified protein MSP1 has no toxic effect on rice leaves [40]. Harpin, one of the classic elicitors, can activate plant defense reaction well-known as HR. Whereas, it is also necessary for the pathogenicity of pathogenic bacteria including *Erwinia*, *Pseudomonas*, and *Xanthomonas* [41]. XEG1, the recently identified effector from the soybean pathogen *Phytophthora sojae*, actions both as a major virulence factor during soybean infection and a pathogen-associated molecular pattern (PAMP) recognized by the plant’s PAMP recognition machinery to induce the downstream defense responses [42]. Similar to XEG1, the plant defense responses induced by MoHrip1 may also belong to the first layer of immunity called PAMP-triggered immunity (PTI), namely, MoHrip1 probably functions as a PAMP. Therefore, with dual function as a PAMP to induce the following PTI and an effector to facilitate fungal infection, MoHrip1 further blurs the distinction between PAMP and effector, PTI and ETI [43].

The defects of *mohrip1*-deleted mutants in pathogenicity generally due to the activated immune system in rice cells. Thus, we estimated the transcripts of some related genes at 3 dpi and found that the expression levels of *PBZ1*, *PAL* and *PR1a* were all significantly higher in rice leaves inoculated with *mohrip1*-deleted mutant than WT, partially illustrating the significance of MoHrip1 in the inactivation of plant immunity (Figure 6). In our previous researches about MoHrip1, when the rice seedlings, either Nipponbare or transgenic lines containing the *mohrip1* gene, are attacked by the blast fungus, the transcription of some pathogenesis-related (PR) genes are all induced. And the induction levels for these genes are even higher when MoHrip1 exists in extra, either expressed in the transgenic rice or sprayed in the purified recombinant form [24,25]. Meanwhile, even applied exogenously alone without the inoculation of fungus, MoHrip1 activates the expression of these defense related genes significantly [39]. These results may seem contradictory to the results obtained in this study, however, the exposing methods of MoHrip1 in these situations are different. When expressed in the transgenic rice or sprayed in the recombinant form, MoHrip1 may exist continuously and distribute widely. While when secreted in the process of *M. oryzae* infecting rice, the protein is only significantly induced in the infection stage and limited to the infection sites. Furthermore, the concentrations of MoHrip1 in these conditions are also different or uncontrollable. Considering all these factors, MoHrip1 may act as a virulence factor or a PAMP in different situations, resulting in the different transcription of these related PR genes.

Contrasting to other defense related genes, the expression of *PR5* in rice was significantly induced by the inoculation of WT (Figure 6). PR5 proteins are a kind of thaumatin-like proteins (TLPs) widely distributing in plants. Their expression is often induced by microbial infection and pathogenic elicitors, osmotic stress, wounding and plant hormones [44]. The antifungal activity of PR5 is well-documented no matter in the form of heterologous expressed protein [45,46] or transgenic plants [47]. Meanwhile, they can mediate the signal transduction by interaction with their receptors or ligands [44]. BcIEB1 is a probable PAMP identified from the broad range phytopathogenic fungus *Botrytis cinerea*. It can interact with osmotin, one of the PR5 proteins in tobacco, and this interaction can protect the fungus from the antifungal activity of osmotin as well as modulate the elicitor activity of BcIEB1 on plants [48]. Both MoHrip1 and Alt a 1, an allergen causing the chronic asthma in children identified from *Alternaria alternata*, belong to the Alt a 1 family. It has been proved that Alt a 1 can interact with the PR5 thaumatin-like proteins from kiwi and further inhibit their antifungal activity [49]. Similar to BcIEB1 and Alt a 1, we assumed that MoHrip1 probably interacts with PR5 proteins and this assumption has been further confirmed by a yeast two-hybrid system in another study [50]. Therefore, the complex interaction between MoHrip1 and PR5 may partly determine the difference in expression of PR5 compared with other surveyed defense related genes.

Furthermore, we also measured the transcription of several genes associated with the biosynthesis of phytoalexins. As secondary metabolites induced by pathogen attack, phytoalexins often possess intense antimicrobial activities. In rice, most identified phytoalexins are diterpenoid compounds [34]. Some genes related to the production of diterpenoid phytoalexins, such as *KSL7*, *KSL10*, *KOL4*, *CYP76M8*, were all markedly improved in the rice plants infected by *mohrip1* deletion mutant (Figure 6). Sakuranetin is the only identified flavonoid phytoalexin in rice so far. Naringenin 7-*O*-methyltransferase coding gene *NOMT* is crucial for the production of sakuranetin by catalyzing the bioconversion of sakuranetin from its precursor [51]. Consistent with our prediction, both the expression of *NOMT* and the content of sakuranetin were all significantly higher in rice leaves inoculated with *mohrip1*-deleted mutant than WT (Figure 7A,B). As for the biosynthesis of diterpenoid phytoalexins, we measured the content of momilactone A as a representative and obtained a similar result as sakuranetin (Figure 7C). Furthermore, we also measured the contents of other defensive compounds, such as total phenols and flavonoids. Their accumulation also demonstrated the same tendency to sakuranetin and momilactone A (Figure 7D,E). Therefore, it is probable that the increased aggregation of defensive compounds partially resulting in the defects of *Δmohrip1* mutants in penetration and proliferation.

In conclusion, this study demonstrated that *M. oryzae* effector gene *mohrip1* was significantly induced during infection. Although insignificant for the growth, conidiation, and formation of appressoria, MoHrip1 was crucial for the penetration and colonization of *M. oryzae* by inactivation of host immune responses. And this was the main cause for the compromised virulence of *Δmohrip1* mutants. Our research results are effective for the elucidation of the unique infection strategy deployed by *M. oryzae* and control of rice blast disease.

## 4. Materials and Methods

### 4.1. Strains and Culture Conditions

*M. oryzae* wild type strain KJ201 (WT) kindly supplied by Professor Wende Liu, *mohrip1*-deleted strains, complementation strain assayed in this study were all cultured on OA plates at 26 °C. To get large amount of conidia for inoculation, induction was made by placing the plates under continuous light for 10 to 15 days [52]. The conidial suspension was prepared by scraping from OA plates with sterile water and filtrating through one layer of Miracloth. While for fungal mycelia, liquid CM was inoculated with fungal plugs and cultured in a shaker at 26 °C with a speed of 180 rpm. The mycelia were harvested after two to three days and used for nucleic acid extraction and protoplast preparation.

### 4.2. QRT-PCR Analysis of Mohrip1 Expression

To analyze the expression mode of *mohrip1* at different developmental and infectious stages of *M. oryzae*, mycelia, conidia and infected leaf samples were all tested. For the obtaining of leaf samples, rice seedlings grown to three-week-old were inoculated with fungal conidia (1 × 10^8^ conidia mL^−1^ in 0.2% Tween 20) according to the classic spraying method [10]. The inoculated plants were placed in a chamber with high humidity at 26 °C for 24 h in dark and then reset the relative humidity and photoperiod respectively to 80% and 12 h light/dark [27]. Leaf samples were collected respectively at 1, 2, 3, 4, and 5 dpi. Total RNA was extracted using the fungal RNA kit (Omega Bio-Tek, Norcross, GA, USA), and then reversely transcribed into cDNA using the EasyScript One-Step gDNA Removal and cDNA Synthesis Supermix (TransGen Biotech, Beijing, China). With the obtained cDNA as template and *M. oryzae Actin* (MGG_03982) as referring gene, qRT-PCR assays were performed on an ABI 7500 real-time PCR system (Applied Biosystems, Foster city, CA, United States) using TransStart Top Green qPCR SuperMix (TransGen Biotech, Beijing, China). This bioassay was conducted three times independently, each time with three repeats. Primers used were listed in Appendix A.

### 4.3. Vector Construction

According to the standard principle of homologous recombination described in the previous research [53], the knockout vector for *mohrip1* was constructed. Briefly, taking the genomic DNA of *M. oryzae* as template, 739-bp upstream and 766-bp downstream flanking regions of *mohrip1* were respectively amplified by PCR. The upstream sequence was digested by *BamH*I and *EcoR*I and fused to the 5′-end of *HPT* gene on the plasmid pKOV21 [54], generating pKOV21-*mohrip1-U*. Then the resultant vector and the downstream sequence were respectively digested by *Hind*III and *Kpn*I and connected together to form the knockout construct pKOV21-*mohrip1-U-D*.

To further verify the function of *mohrip1*, gene complementation was conducted by transformation of *mohrip1*-deleted mutant with the complementation vector pYK11-*mohrip1*, which was constructed by subcloning the entire coding region of *mohrip1* with its native promoter and terminator (~2.4 kb) into the multiple cloning site on pYK11. Primers used were listed in Appendix A.

### 4.4. Fungal Transformation and Confirmation

PEG-mediated protoplast transformation was adopted in this study [55]. The *mohrip1*-deleted mutants were generated by transforming the construct pKOV21-*mohrip1-U-D* into WT and screened by resistance to hygromycin B. In order to construct the complementation mutants, the plasmid pYK11-*mohrip1* was introduced to protoplasts released from the *mohrip1*-deleted strain. The putative mutants were selected by resistance to zeocin. All putative mutants in the section of *mohrip1* deletion and complementation were confirmed by PCR and Southern Blot [54]. Primers for PCR and probe construction were listed in Appendix A.

### 4.5. Phenotypic Characterization

The developmental traits were investigated according to the previous methods with a little modification [56,57]. For the vegetative growth and colony morphology, mycelial blocks of 5 mm in diameter were picked out from ten-day-old OA plates and inoculated on fresh CM and PDA plates. After incubation at 26 °C in dark for five days, the radial growth was measured with a vernier caliper. For the measurement of conidiation, mycelial plugs of 5 mm × 5 mm were put on glass slides and incubated in a moist container. The conidia on conidiophores were observed under a microscope at 48 hpi. The quantity of conidia produced on OA plate was estimated by rubbing the plate with 5 mL of sterilized distilled water and counting with a hemacytometer [56]. For conidial germination and formation of appressoria, conidial suspensions (1 × 10^5^ conidia mL^−1^) of 20 µL were dropped on coverslips and placed in a humid box. At least 100 conidia for each strain were examined microscopically respectively at 2, 6 and 12 hpi [56]. All bioassays were conducted at least three times, each time with three to five repeats.

### 4.6. Pathogenicity Assays

In the pathogenicity assays, rice cultivar Nipponbare and conidial suspensions with a same concentration of 2 × 10^5^ conidia mL^−1^ in 0.2% Tween 20 were adopted. In the spraying inoculation, fungal conidia were quantitatively and uniformly sprayed on three-week-old rice seedlings. The inoculated plants were placed in dark for 24h and then returned to the normal growth conditions [10,27]. After seven days, the lesion progressions were observed according to the six types divided by Valent et al. [26]. Meanwhile, the disease index was also investigated referring to the international specification for rice blast disease [24]. In addition, the relative expression levels of certain defense related genes at 3 dpi were also investigated by qRT-PCR after normalization with rice *β-Actin* gene (Os11G0163100).

For the punch inoculation assays, 6 to 8-week-old rice plants were used. According to the method of Liu et al. [27], conidial suspension of 10 µl was dropped to the wound area on rice leaf made by a puncher. The wounded area with conidial suspension was sealed with a piece of transparent film, and then transferred into the growth chamber. After 12 days, leaf samples of 3 cm in length containing the lesions were harvested and photographed. The images were analyzed with the Adobe Photoshop software to calculate the relative lesion area. Moreover, the relative fungal growth, represented by the ratio of C_T_ values of *Pot2* gene (MGG_13294.6) in *M. oryzae* versus the *Ubiquitin* gene (Os03g13170) in rice, was determined by DNA-based qPCR. The relative fungal biomass was then calculated according to the equation 2 ^CT(*Ubiquitin*) − CT(*Pot2*)^ [10].

These experiments were all repeated at least three times, each time with three repeats, and the representative results from one time were displayed. Primers used were listed in Appendix A.

### 4.7. Rice Leaf Sheath Infection Assays

Rice leaf sheath inoculation assays were conducted to observe the entire infection process of *M. oryzae* [6,58]. Leaf sheaths were excised from the 1.5 to 2.5-month-old rice plants and inoculated with a conidial suspension of 5 × 10^4^ conidia mL^−1^ in 0.2% Tween 20. The inoculated sheaths were observed microscopically at different time points post inoculation. According to the four infection levels (Level 0, just appressoria without penetration; Level 1, with penetration peg or primary IH; Level 2, with sparse secondary IH restricted in the first infected rice cell; Level 3, with abundant IH in the initial and even neighboring rice cells) we divided in another research [30], the infection status was investigated. For each sample, at least 100 penetrated sites were observed. Three assays were conducted independently, each time with three to five repeats.

### 4.8. Quantification of Defensive Compounds

The measurement of phytoalexins including sakuranetin and momilactone A was mainly according to the method of Nishimura et al. [9] by HPLC-MS/MS. Briefly, rice leaves inoculated with different *M. oryzae* strains were respectively collected at 3 dpi. Thirty-five milligrams of leaf samples were grinded into powder and incubated in 1 mL extraction buffer methanol:H_2_O (1:1, *v*/*v*) for 30 min. Then two microliters of the supernatants were subjected to HPLC-MS/MS. For the estimation of total phenols and flavonoids, leaf samples with the same weight were harvested, cut into 1 to 2 mm sections and immersed in methanol: HCl (99:1, *v*/*v*) at 4 °C. After 24 h, the absorbance values of OD_280 nm_ and OD_325 nm_, respectively representing the contents of total phenols and flavonoids, were measured using SpectraMax 190 [59]. Three times were conducted for the assays, each time with three to five repeats.

### 4.9. Statistics

All bioassays were conducted three independent times, each time with at least three repeats. Statistical analysis was performed using the software of SAS System for Windows V8. Data from independent biological replicates were analyzed with ANOVA (analysis of variance). Significance analysis was carried out using the Duncan’s multiple range test.

## Figures and Tables

**Figure 1 ijms-20-01643-f001:**
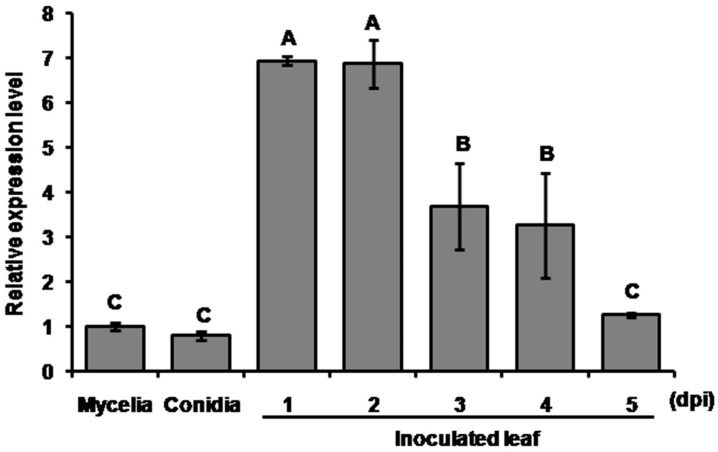
QRT-PCR analysis of *mohrip1* expression in *M. oryzae* at different developmental and infective stages. Transcription of *mohrip1* was significantly induced in the first 2 dpi. The relative expression levels of *mohrip1* on the vertical axis were relative to that in the axenic mycelia after normalization with *M. oryzae Actin* gene. Values are the means of three independent biological replications. The error bars represent standard deviation (SD). “A”, “B” and “C” indicate the significance analyzed using Duncan’s multiple range test at *p* < 0.01. dpi, days post inoculation.

**Figure 2 ijms-20-01643-f002:**
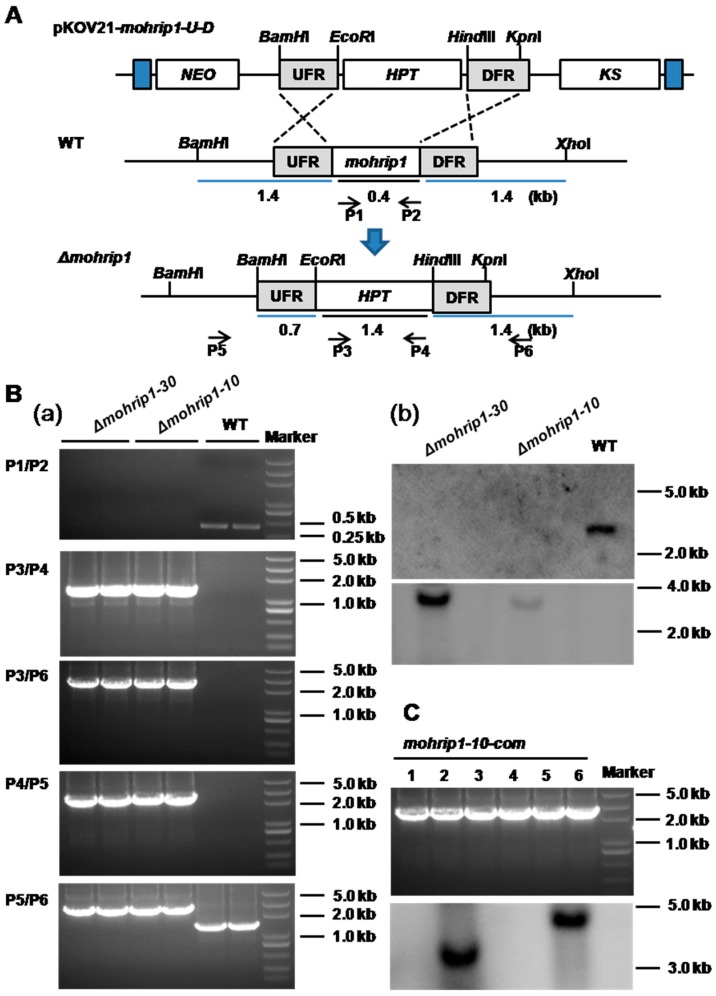
Construction and confirmation of *mohrip1* deletion and complementation mutants. (**A**) Deletion of *mohrip1* according to homologous recombination. pKOV21-*mohrip1-U-D*, knockout vector for *mohrip1*; WT, pattern of *mohrip1* in the genome of *M. oryzae*; *Δmohrip1*, deletion mutants containing *HPT* gene replacing *mohrip1*; UFR, upstream flanking region of *mohrip1*; DFR, downstream flanking region of *mohrip1*. Values represent the size of corresponding fragments. P1 to P6 were six primers used for the mutant confirmation by PCR. (**B**) Confirmation of the deletion mutants. (**a**) PCR confirmation; (**b**) Southern Blot validation using *mohrip1* (**upper**) and *HPT* (**lower**) probes; (**C**) Confirmation of several independent complementation strains by PCR (**upper**) and Southern Blot (**lower**) with *mohrip1* probe. The 1—6 were six complementation strains confirmed by PCR, from which two strains were chosen for Southern Blot. One strain named *mohrip1-10-com* was randomly selected for the subsequent bioassays. *Δmohrip1-10*, *Δmohrip1-30*, *mohrip1-10-com* and WT respectively represent the two deletion mutants, one complementation mutant and the wild type strain.

**Figure 3 ijms-20-01643-f003:**
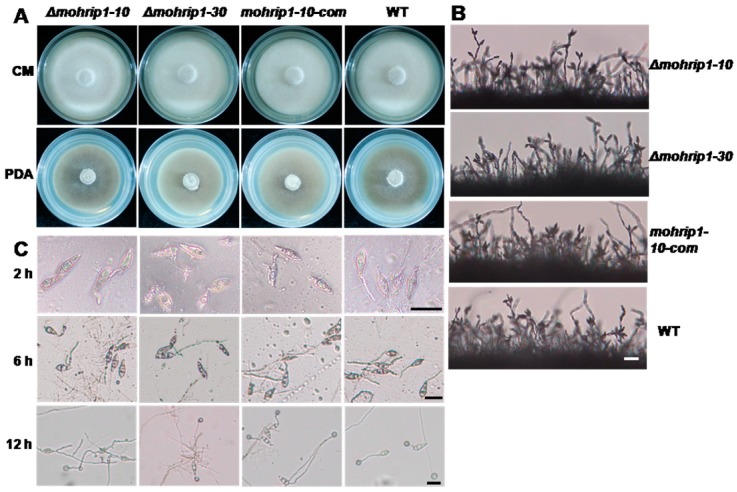
Characterization of *mohrip1* deletion and complementation mutants. MoHrip1 was not required for *M. oryzae* saprophytic growth and asexual reproduction. (**A**) Colony growth of *M. oryzae* strains on CM and PDA plates after incubation for five days. (**B**) Conidiation of *M. oryzae* strains. (**C**) Conidial germination and appressorial formation of *M. oryzae* strains. Observation was respectively conducted at 2, 6 and 12 hpi. Scale bar = 20 µm.

**Figure 4 ijms-20-01643-f004:**
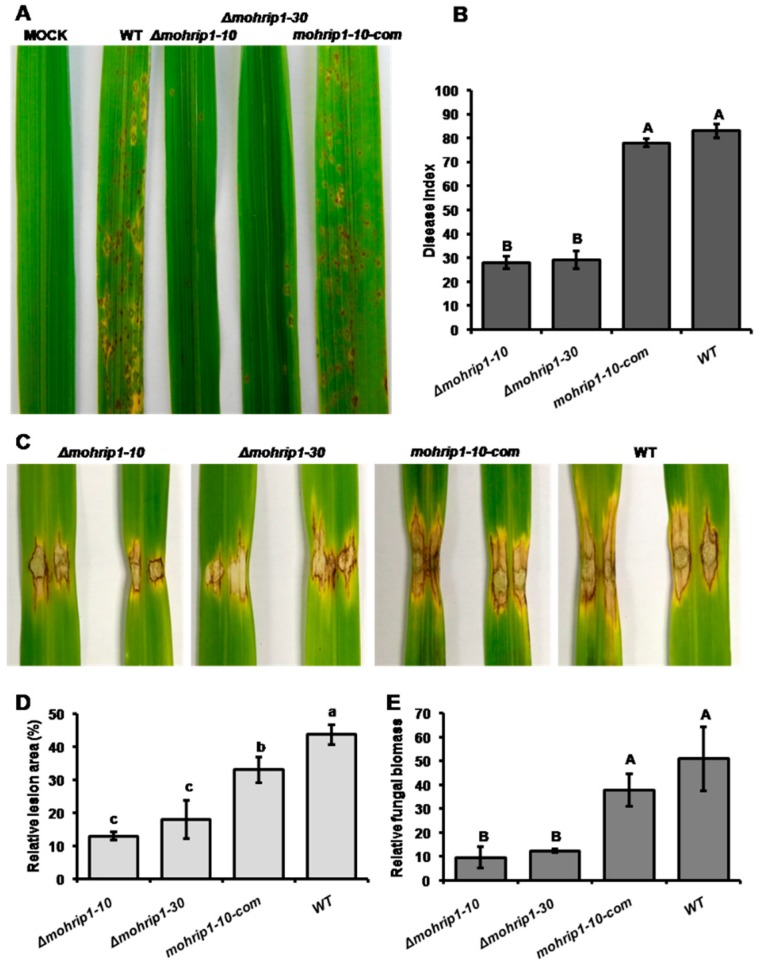
Pathogenicity of *Δmohrip1* mutants. The *mohrip1* deletion mutants were significantly impaired in pathogenicity. (**A**) Virulence bioassay on rice seedlings conducted by the spraying inoculation method. The three-week-old rice seedlings were used for the tests. The concentrations of conidial suspensions were all adjusted to 2 × 10^5^ conidia mL^−1^ in 0.2% Tween 20. MOCK, sterile water added with 0.2% Tween 20 taken as a negative control. At 7 dpi, the symptoms of rice plants treated by different strains were investigated. (**B**) Disease index of rice blast in spraying inoculation bioassay. Disease indexes were investigated according to the ten grades of international specification. Data are representative of three independent assays, each assay with three repeats. Error bars represent SD. “A” and “B” indicate the significance analyzed using Duncan’s multiple range test at *p* < 0.01. (**C**) Virulence bioassay on rice plants conducted by the punch inoculation method. 6 to 8-week-old rice leaves were wounded by a puncher. 10 µl of conidial suspensions (2 × 10^5^ conidia ml^−1^ in 0.2% Tween 20) were respectively dropped on the wound areas, sealed with transparent film and incubated at 26 °C. After 12 days, the disease status was investigated and photographed. (**D**) Relative lesion area in punch inoculation bioassay. Rice leaf samples were harvested by cutting 3 cm long sections containing the lesions. Measurement was conducted using the Adobe Photoshop software. Error bars represent SD. The “a”, “b” and “c” respectively represent the significance analyzed using Duncan’s multiple range test at *p* < 0.05. (**E**) Relative fungal biomass in punch inoculation bioassay. DNAs of the harvested 3 cm leaf sections containing the lesions were extracted according to the CTAB method. Representing the relative fungal biomass, the value 2 ^[CT(*OsUbiquitin*) − CT(*MoPot2*)]^ was calculated according to the result of DNA-based qPCR. Values are the means of three independent biological replications, and error bars represent SD. “A” and “B” indicate the significance analyzed using Duncan’s multiple range test at *p* < 0.01. *Δmohrip1-10*, *Δmohrip1-30*, *mohrip1-10-com* and WT respectively represent the two deletion mutants, one complementation mutant and the wild type strain.

**Figure 5 ijms-20-01643-f005:**
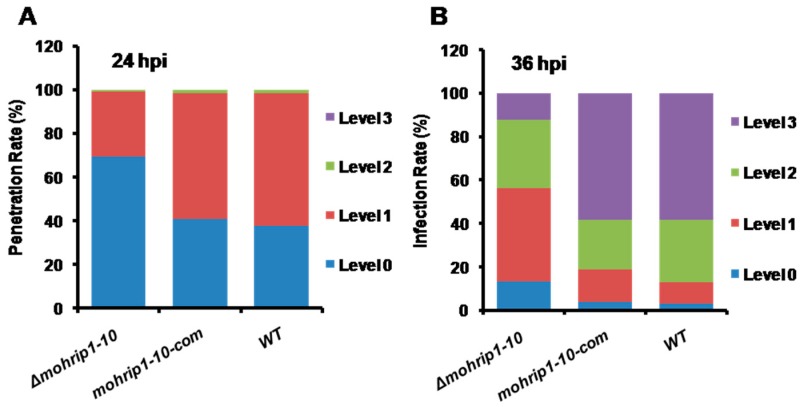
MoHrip1 is important for *M. oryzae* penetration and proliferation. According to the four levels (Level 0, just appressoria without penetration; Level 1, with penetration peg or primary IH; Level 2, with sparse secondary IH restricted in the first infected rice cell; Level 3, with abundant IH in the initial and even neighboring rice cells) we divided in another research, the infection status of rice leaf sheaths inoculated with *M. oryzae* strains (5 × 10^4^ conidia ml^−1^ in 0.2% Tween 20) was respectively investigated at 24 hpi and 36 hpi. (**A**) Statistical analysis of the penetration rates of *M. oryzae* strains at 24 hpi. (**B**) Statistical analysis of the infection rates of *M. oryzae* strains at 36 hpi. Data are representative of three independent assays, each assay with three repeats. For each repeat, at least 100 appressoria were observed. *Δmohrip1-10*, *mohrip1-10-com* and WT respectively represent one deletion mutant, one complementation mutant and the wild type strain.

**Figure 6 ijms-20-01643-f006:**
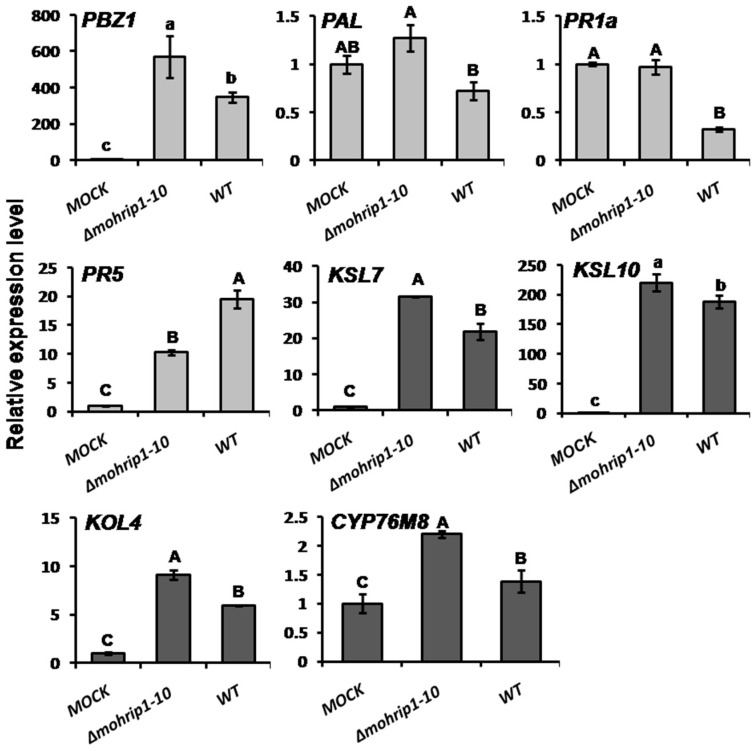
MoHrip1 suppresses the transcription of certain defense related genes in rice. Transcription levels of these genes were induced stronger in rice infected with *Δmohrip1-10* than WT at 3 dpi other than *PR5*. The vertical axis indicates the expression levels relative to those in the water sprayed leaves (MOCK) after normalization with the rice *β-Actin* gene (Os11G0163100). Gray columns represent the PR or the defense pathway related genes, while brown ones represent genes taking part in the biosynthesis of phytoalexins. Three independent biological experiments were conducted, each with three replicates. Error bars represent SD. The “a”, “b”, “c” and “A”, “B”, “C” respectively represent the significance analyzed using Duncan’s multiple range test at *p* < 0.05 and *p* < 0.01. MOCK, *Δmohrip1-10*, and WT respectively represent the rice seedlings treated with 0.2% Tween 20, *mohrip1* deletion mutant and the wild type strain.

**Figure 7 ijms-20-01643-f007:**
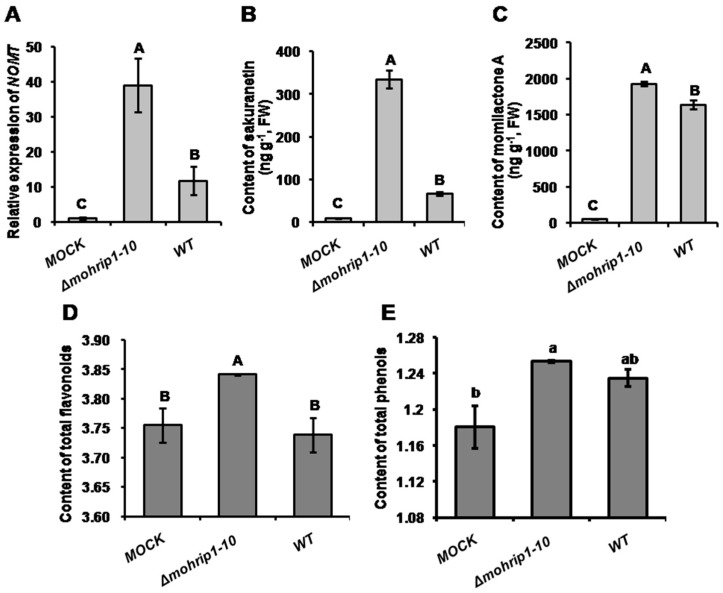
MoHrip1 inhibits the production of defensive compounds. (**A**) Improved transcription of *OsNOMT* by the treatment with *Δmohrip1-10* at 3 dpi. *OsNOMT* encodes naringenin 7-*O*-methyltransferase critical for the biosynthesis of sakuranetin, the only known flavonoid phytoalexin in rice to date. (**B**) Contents of sakuranetin measured by HPLC-MS/MS. (**C**) Contents of momilactone A measured by HPLC-MS/MS. (**D**) Contents of total flavonoids measured by spectrophotometer. (**E**) Contents of total phenols measured by spectrophotometer. The contents of total flavonoids and phenols were respectively indicated by the absorbance values at OD_280 nm_ (for total phenols) and OD_325 nm_ (for total flavonoids). The bioassay conducted three times independently, each time with three repeats. Error bars represent SD. The “a”, “b” and “A”, “B”, “C” respectively represent the significance analyzed using Duncan’s multiple range test at *p* < 0.05 and *p* < 0.01. MOCK, *Δmohrip1-10*, and WT respectively represent the rice seedlings treated with 0.2% Tween 20, *mohrip1* deletion mutant and the wild type strain.

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
