# Peer review of "The Secreted Protein MoHrip1 Is Necessary for the Virulence of Magnaporthe oryzae"

_ijms, 2019, doi:10.3390/ijms20071643_

Round 1
Reviewer 1 Report
Dear Author,
Thank you very much for choosing the renowned journal ‘Internal Journal of Molecular Sciences’ for submitting article titled ‘The Secreted Protein MoHrip1 is Necessary for the Virulence of Magnaporthe oryzae’. The article focuses on the very important fungal pathogen Magnaporthe oryzae, its important effectors, fungal infection and disease development.
By the way, there are very few suggestions to improve the article. Please check the following points-
L360-from where the wild strain comes from and from which plant/which country?
L363- Is this continuous light? Or 12h-12h light-dark condition? Please confirm
L370- As far I know it’s not easy to produce spore suspension in a conc. of 1 × 108 spores/ml. could you please confirm where the sporulation method was right? I have seen in many places of the article the author used spore suspension in different concentrations. Eg. L370, L415, L436 etc.
When use company name for any reagent, follow the international standard to write like company name, city & country. Eg-L374 (is this Omega or promega?), L376 (name, city, country), L378
Minor grammatical errors were observed, eg. L31 causal agent instead of cause agent
The author/s could add the supplementary photo to the main article.
Thank you in advance
Author Response
Dear Editor and Reviewer,
Thank you very much for your letter and for the reviewer’s comments and suggestions concerning our manuscript entitled " The Secreted Protein MoHrip1 is Necessary for the Virulence of Magnaporthe oryzae" (ID: ijms-434641). Those comments are all valuable and helpful for revising and improving our manuscript. We studied those comments carefully and made revisions following the reviewer’s suggestions. Revised portions are highlighted using the "Track Changes" function in the revised manuscript. The main revisions in the manuscript and the responses to the reviewer’s comments are as following:
Point 1: L360-from where the wild strain comes from and from which plant/which country?
Response 1: The wild type strain KJ201 was kindly supplied by Professor Wende Liu of Institute of Plant Protection, Chinese Academy of Agricultural Sciences. His team did some researches using this strain, such as a paper entitled “Identification and Characterization of In planta–Expressed Secreted Effector Proteins from Magnaporthe oryzae That Induce Cell Death in Rice” in Molecular Plant-Microbe Interactions (MPMI Vol. 26, No. 2, 2013, pp. 191–202. http://dx.doi.org/10.1094/MPMI -05-12-0117-R.). We added the source of this strain in the section of Materials and Methods and Acknowledgments in the revised manuscript.
Point 2: L363- Is this continuous light? Or 12h-12h light-dark condition? Please confirm
Response 2: We confirmed that the plates were placed under the continuous light.
Point 3: L370- As far I know it’s not easy to produce spore suspension in a conc. of 1 × 108 spores/ml. could you please confirm where the sporulation method was right? I have seen in many places of the article the author used spore suspension in different concentrations. Eg. L370, L415, L436 etc.
Response 3: We confirmed that the concentration adopted in the quantification of mohrip1 expression was 1 × 108 conidia ml−1 (L390 in revised manuscript). We thought that the relative high inoculation concentration could make the detection of mohrip1 expression easier. And this concentration was determined according to a study titled “MoTup1 Is Required for Growth, Conidiogenesis, and Pathogenicity of Magnaporthe oryzae” in Molecular Plant Pathology (Vol. 16, No. 8, 2015, pp. 799–810. DOI: 10.1111/mpp.12235). However, for the concentrations in L436 (2 × 105 conidia ml-1) and L458 (5 × 104 conidia ml−1), they were respectively selected for the normal inoculation and easy observation.
Point 4: When use company name for any reagent, follow the international standard to write like company name, city & country. Eg-L374 (is this Omega or promega?), L376 (name, city, country), L378
Response 4: According to your kind suggestion, we corrected the company names following the international standard in the revised manuscript. The company name appeared in L394 was Omega Bio-Tek, Norcross, GA, United States. L396, Transgen Biotech, Beijing, China. L399, Applied Biosystems, Foster city, CA, United States.
Point 5: Minor grammatical errors were observed, eg. L31 causal agent instead of cause agent
Response 5: We carefully checked the grammatical errors and edited the language and style in the revised manuscript. The “cause agent” in L31 was changed to “causal agent”.
Point 6: The author/s could add the supplementary photo to the main article.
Response 6: Following your kind suggestion, we added the supplementary photo to the main article as Figure 3 in the revised manuscript.
Reviewer 2 Report
Manuscript ijms-434641: Referees Report
Within their manuscript the authors aim to characterize proteins or peptides secreted by the plant pathogenic fungus Magnaporthe oryzae. In general the work is without any doubt of interest to the scientific community the field of plant-pathogen interaction research. However, considering previous work published by the authors I concluded that the novelty in the present manuscript is down to qPCR studies, showing that transcript levels of genes related to plant defence are down regulated in a significant manner. I did wonder whether the reduced virulence of mutant strains is a novel finding or whether this was the basis of previous publications.
As mentioned before quantitaive PCR assesses the transcript level of a gene of putative relevance. As a general comment to the authors I would strongly recommend carrying out proteome analysis which would back hypothesis raised as a consequence to indications from qPCR experiments.
A second general comment to the authors would be to re-phrase some of the statements. E.g. in the first sentence of the introduction the authors state that “… plant pathogens of biotrophic or hemibiotrophic fungi feed on living tissue for the whole life cycle…”. First of all "plant pathogens of fungi" requires re-phrasing. Secondly it appears remarkable that spores germinate and differentiate outside the plant, respectively outside the host tissue. Apart from that Magnaporthe oryzae is to my knowledge perfectly able to survive on other or even artificial nutrient sources.
The manuscript is of interest whereas I feel that major revisions are required prior to publication:
The molecular biology section: Page 4, Fig.2:
The molecular manipulation of the generated mutants and especially the detection is not properly documented. The construction and/or verification of the complemented strain is completely missing in Fig.2, although it is mentioned in the text and the caption of Fig 2. The picture of the southern blot is of very poor quality and the shown section is much too small, in addition the marker is lacking. As far as I understood, the authors used the gene mohrip1 as probe in the southern analysis. This is not sufficient to demonstrate a clear and clean evidence for the directed molecular manipulation by replacing the gene with the resistance gene HPT. In the case of mutant strains there is simply no signal to be seen here, that is too thin to be a proof. A southern blot should be planned and performed in such a way that you can see different signals in the wt and the mutant and the complementary strain.
Virulence and pathogenicity assays
In my experience virulence and pathogenicity assays can be subject to significant variations. I would strongly recommend to include assays with well-established non-pathogenic mutant strains available in the scientific community as controls.
Quantification of defensive compounds:
The description of methods is rather vague. Sakuranetin was apparently measured by HPLC-MS. This requires further description. Futhermore I am worried about the quantification of total phenols and flavonoids. Does this imply that all flavenoids or phenols are phytoalexins? I would strongly recommend further analysis concerning the phytoalexins. Otherwise the basis for a hypothesis concerning the reduced levels of plant defence metabolites as a reason for succesful plant-pathogen-interaction is not given or too vague. Further experiments are required.
Relative fungal biomass
This needs re-writing and further explanation.
Author Response
Dear Editor and Reviewer,
Thanks a lot for your letter and for the reviewer’s comments and suggestions concerning our article titled " The Secreted Protein MoHrip1 is Necessary for the Virulence of Magnaporthe oryzae" (ID: ijms-434641). Those comments are all valuable and meaningful for the revision of our manuscript. We responded those comments in a point-by-point way and made revisions following the reviewer’s suggestions. Revised portions are highlighted using the "Track Changes" function in Microsoft Word in the revised manuscript. The main revisions in the manuscript and the responses to the reviewer’s comments are listed as follows:
Point 1: Within their manuscript the authors aim to characterize proteins or peptides secreted by the plant pathogenic fungus Magnaporthe oryzae. In general the work is without any doubt of interest to the scientific community the field of plant-pathogen interaction research. However, considering previous work published by the authors I concluded that the novelty in the present manuscript is down to qPCR studies, showing that transcript levels of genes related to plant defence are down regulated in a significant manner. I did wonder whether the reduced virulence of mutant strains is a novel finding or whether this was the basis of previous publications.
Response 1: Thank you very much for your comments. Just as you know, we had published some papers about the protein MoHrip1. However, whether in the form of recombinant protein expressed in E. coli [1,2] or transformed into plants as a exogenous gene [3,4], MoHrip1 was all studied as an independent protein almost without considering of its source fungus Magnaporthe oryzae. Although these publications proved the function of MoHrip1 as a protein elicitor to activate plant immunity, its intrinsic biological function in M. oryzae was still elusive. In the study, we were mainly to solve this problem by gene deletion and complementation. Therefore, the function of MoHrip1 as a virulence factor in M. oryzae was a new finding rather than the basis of previous publications.
[1] Chen M, Zeng H, Qiu D, Guo L, Yang X, et al. (2012) Purification and Characterization of a Novel Hypersensitive Response-Inducing Elicitor from Magnaporthe oryzae that Triggers Defense Response in Rice. PLoS ONE 7(5): e37654. doi:10.1371/journal.pone.0037654
[2] Lv S, Wang Z, Yang X, Guo L, Qiu D and Zeng H (2016) Transcriptional Profiling of Rice Treated with MoHrip1 Reveal the Function of Protein Elicitor in Enhancement of Disease Resistance and Plant Growth. Front. Plant Sci. 7:1818. doi: 10.3389/fpls.2016.01818
[3] Wang Z, Han Q, Zi Q, Lv S, Qiu D, Zeng H (2017) Enhanced disease resistance and drought tolerance in transgenic rice plants overexpressing protein elicitors from Magnaporthe oryzae. PLoS ONE 12(4): e0175734. https://doi.org/10.1371/journal.pone.0175734
[4] Han Q, Wang Z, He Y, Xiong Y, Lv S, Li S, Zhang Z, Qiu D and Zeng H (2017) Transgenic Cotton Plants Expressing the HaHR3 Gene Conferred Enhanced Resistance to Helicoverpa armigera and Improved Cotton Yield. International Journal of Molecular Sciences, 18, 1874, doi:10.3390/ijms18091874
Point 2: As mentioned before quantitative PCR assesses the transcript level of a gene of putative relevance. As a general comment to the authors I would strongly recommend carrying out proteome analysis which would back hypothesis raised as a consequence to indications from qPCR experiments.
Response 2: Thank you again for your kind suggestion. The suggested proteome analysis might be helpful in backing the results of qPCR, however, this portion was not the main body of our study. As revealed by the title, our fundamental purpose was to illustrate the native function of MoHrip1 in M. oryzae. By the construction of deletion and complementation mutants, we found the nature of MoHrip1 as a virulence factor. As for the qPCR assay, it was just used to partially explain the reduced virulence of mohrip1-deleted mutants and the role of MoHrip1 in the inactivation of plant immunity during plant-pathogen interaction. Therefore, the proteome analysis might be dispensable.
Point 3: A second general comment to the authors would be to re-phrase some of the statements. E.g. in the first sentence of the introduction the authors state that “… plant pathogens of biotrophic or hemibiotrophic fungi feed on living tissue for the whole life cycle…”. First of all "plant pathogens of fungi" requires re-phrasing. Secondly it appears remarkable that spores germinate and differentiate outside the plant, respectively outside the host tissue. Apart from that Magnaporthe oryzae is to my knowledge perfectly able to survive on other or even artificial nutrient sources.
Response 3: Thanks for your comments. We carefully edited the language and style in the revised manuscript. Firstly, we rephrased the first sentence in Introduction to “Biotrophic or hemibiotrophic pathogenic fungi often feed on living host tissues for the whole life cycle or early infection stages”. Secondly, as for your suggestion about M. oryzae, its spores do germinate and differentiate outside the plant. As we described in the second paragraph in a typical disease cycle, the infections initiate when the spores land on the surface of rice leaves, a place outside the plant where the spores germinate and differentiate. Besides, M. oryzae can survive on other plants or even artificial nutrient sources and we cultured this fungus using artificial medium in this study. However, during its infective interaction with plant, it is a well-known hemibiotrophic fungus and has a stage feeding on living plant tissue.
Point 4: The molecular biology section: Page 4, Fig.2:
The molecular manipulation of the generated mutants and especially the detection is not properly documented. The construction and/or verification of the complemented strain is completely missing in Fig.2, although it is mentioned in the text and the caption of Fig 2. The picture of the southern blot is of very poor quality and the shown section is much too small, in addition the marker is lacking. As far as I understood, the authors used the gene mohrip1 as probe in the southern analysis. This is not sufficient to demonstrate a clear and clean evidence for the directed molecular manipulation by replacing the gene with the resistance gene HPT. In the case of mutant strains there is simply no signal to be seen here, that is too thin to be a proof. A southern blot should be planned and performed in such a way that you can see different signals in the wt and the mutant and the complementary strain.
Response 4: Thank you very much for the suggestions about experimental details which are important for the improvement of our figure and article. Just as we described in the Materials and Methods, the construction of complementation mutant was conducted by subcloning the entire coding region of mohrip1 with its native promoter and terminator (~2.4 kb) into the multiple cloning site on pYK11 and then transforming this vector into the deletion mutant. This procedure was easy to understand and often not appearing in the figure, such as literature 5 and 6 listed as follows [5,6].
In the experiment of Southern Blot, we didn’t use the specific probe to detect the marker bands. But by comparing the pictures before and after blotting, we determined the approximate size of target band and signed it on the right side of the picture.
Other than mohrip1 gene functioning as probe, we also analyzed the deletion mutants and WT using HPT gene as probe. Contrasting to the Southern Blot result using mohrip1 probe, the bands respectively appeared in the two deletion mutants. Whereas for WT, no signal was detected. We added this picture to Figure 2B(b), under the blotting picture using mohrip1 probe. Using these two probes to verify the deletion mutant is common, such as literature 5 and 6 [5,6].
[5] Chen Y, Le X, Sun Y, Li M, Zhang H, Tan X, Zhang D, Liu Y and Zhang Z (2017) MoYcp4 is required for growth, conidiogenesis, and pathogenicity in Magnaporthe oryzae. Molecular Plant Pathology, 18(7): 1001–1011. DOI: 10.1111/mpp.12455
[6] Ding S, Liu W, Iliuk A, Ribot C, Vallet J, Tao A, Wang Y, Lebrun M and Xu J (2010) The Tig1 Histone Deacetylase Complex Regulates Infectious Growth in the Rice Blast Fungus Magnaporthe oryzae. The Plant Cell, Vol. 22: 2495–2508. www.plantcell.org/cgi/doi/10.1105/tpc.110.074302
Point 5: Virulence and pathogenicity assays
In my experience virulence and pathogenicity assays can be subject to significant variations. I would strongly recommend to include assays with well-established non-pathogenic mutant strains available in the scientific community as controls.
Response 5: In order to minimize the variations in pathogenicity assays, repetition was conducted with at least three times. In this study, we mainly wanted to compare the difference between WT and the mutants in virulence. Therefore, another non-pathogenic mutant strain might be dispensable. Similarly, another study about protein Rbf1 derived from M. oryzae also doesn’t include other non-pathogenic mutants in the inoculation assay [7].
[7] Nishimura T, Mochizuki S, Ishii-Minami N, Fujisawa Y, Kawahara Y, Yoshida Y, et al. (2016) Magnaporthe oryzae Glycine-Rich Secretion Protein, Rbf1 Critically Participates in Pathogenicity through the Focal Formation of the Biotrophic Interfacial Complex. PLoS Pathog 12(10): e1005921. doi:10.1371/journal.ppat.1005921
Point 6: Quantification of defensive compounds:
The description of methods is rather vague. Sakuranetin was apparently measured by HPLC-MS. This requires further description. Futhermore I am worried about the quantification of total phenols and flavonoids. Does this imply that all flavenoids or phenols are phytoalexins? I would strongly recommend further analysis concerning the phytoalexins. Otherwise the basis for a hypothesis concerning the reduced levels of plant defence metabolites as a reason for succesful plant-pathogen-interaction is not given or too vague. Further experiments are required.
Response 6: According to your kind suggestions, we further detailed the quantification method of phytoalexins including sakuranetin and momilactone A in the revised manuscript as follows:
The measurement of phytoalexins including sakuranetin and momilactone A was mainly according to the method of Nishimura et al. [9] by HPLC-MS/MS. Briefly, rice leaves inoculated with different M. oryzae strains were respectively collected at 3 dpi. Thirty-five milligrams of leaf samples were grinded into powder and incubated in 1 mL extraction buffer methanol:H2O (1:1, v/v) for 30 min. Then two microliters of the supernatants were subjected to HPLC-MS/MS.
We measured the contents of total phenols and flavonoids and took them as indexes of certain defensive compounds in rice plant, not phytoalexins.
Besides, we further measured the content of another phytoalexin momilactone A in rice leaves inoculated with M. oryzae strains. As shown in Figure 7C, consistent to sakuranetin, the deletion mutant also induced significantly higher accumulation of momilactone A.
Point 7: Relative fungal biomass
This needs re-writing and further explanation.
Response 7: The calculating method of relative fungal biomass was further detailed by the addition of an equation 2CT(Ubiquitin) - CT(Pot2) in the Materials and Methods. The description was revised as follows:
Moreover, the relative fungal growth, represented by the ratio of CT values of Pot2 gene (MGG_13294.6) in M. oryzae versus the Ubiquitin gene (Os03g13170) in rice, was determined by DNA-based qPCR. The relative fungal biomass was then calculated according to the equation 2CT(Ubiquitin) - CT(Pot2).